# TidyBot++: An Open-Source Holonomic Mobile Manipulator for Robot Learning

**Jimmy Wu[1], William Chong[2], Robert Holmberg[3], Aaditya Prasad[2], Yihuai Gao[2], Oussama Khatib[2], Shuran Song[2], Szymon Rusinkiewicz[1], Jeannette Bohg[2]**
[1]Princeton University     [2]Stanford University     [3]Dexterity

http://tidybot2.github.io

**Abstract:** Exploiting the promise of recent advances in imitation learning for mobile manipulation will require the collection of large numbers of human-guided demonstrations. This paper proposes an open-source design for an inexpensive, robust, and flexible mobile manipulator that can support arbitrary arms, enabling a wide range of real-world household mobile manipulation tasks. Crucially, our design uses powered casters to enable the mobile base to be fully *holonomic*, able to control all planar degrees of freedom independently and simultaneously. This feature makes the base more maneuverable and simplifies many mobile manipulation tasks, eliminating the kinematic constraints that create complex and time-consuming motions in nonholonomic bases. We equip our robot with an intuitive mobile phone teleoperation interface to enable easy data acquisition for imitation learning. In our experiments, we use this interface to collect data and show that the resulting learned policies can successfully perform a variety of common household mobile manipulation tasks.

**Keywords:** mobile manipulation, imitation learning, holonomic drive

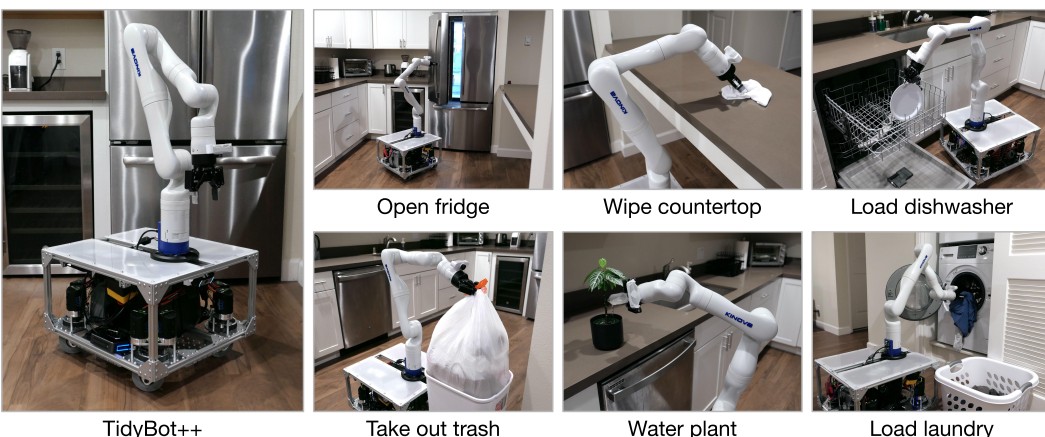

Figure 1: We develop an open-source mobile manipulator with a holonomic base (left), and show that it can perform a variety of household tasks in a real apartment home (right).

## 1   Introduction

Imitation learning from real-world data is starting to show very promising results in robotics for both fixed-arm robots [1, 2, 3, 4, 5, 6, 7] and mobile manipulators [8, 9, 10, 11, 12, 13, 14, 15, 16, 17, 18, 19, 20, 21]. However, one key bottleneck is the availability of data. Unlike in natural language processing, which can train on readily-available data from the internet, real-world data for training robot policies is much harder to come by. As a result, scaling data collection of robotic tasks has

8th Conference on Robot Learning (CoRL 2024), Munich, Germany.

become a high-interest research direction. Recent efforts have collected large robot learning datasets to address this [22, 23, 24, 25, 26, 27]. These datasets were largely collected on fixed-arm robot setups. However, to bring robots to their full potential, mobility is important as it will enable many more tasks in realistic household settings [8].

We believe that one reason there are so few data collection and learning efforts in mobile manipulation is the lack of suitable research hardware. Existing commercial options for mobile bases are often tailored towards industrial or warehouse use cases, and may be ill-suited for household environments due to their large size. They are also often expensive and are typically subject to kinematic constraints.

In this work, we propose an open-source design for a mobile base designed to carry a robot arm sized for use in household environments. In addition to being inexpensive, flexible, and easy to assemble, our base is *holonomic*: able to independently move in any of the three degrees of freedom (DoFs) on the ground plane—$(x, y, \theta)$—at any time. We argue that this is an important advantage for more intuitive teleoperation, and greatly increases the ease of acquiring large amounts of training data for real-world imitation learning.

Nonholonomic robots, such as differential drive (wheelchair-like) or Ackermann drive (car-like) platforms, have constrained degrees of freedom in their motion. The most notable consequence of this is that they cannot move sideways. For example, cars cannot directly drive sideways into a street-side parking spot and must execute a multi-step parallel-parking maneuver.

In contrast, holonomic robots have no kinematic constraints and can simultaneously and independently control all three degrees of freedom. An example of a holonomic vehicle common in everyday life is an office chair, which can be smoothly pushed or rotated in arbitrary directions. This is enabled by the design of the caster wheels (Fig. 2), which have an offset between the vertical axis of the swivel mechanism and the roll axis of the wheel. This offset is a crucial design feature of casters and is what

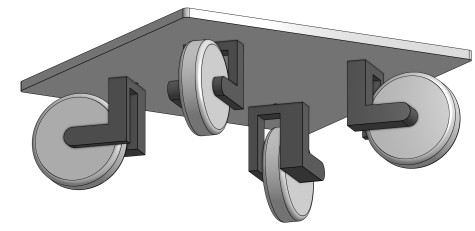

Figure 2: A simplified illustration of caster wheels on a holonomic base.

makes the office chair fully holonomic. It creates a lever arm that causes the wheel to trail behind the vertical axis of the swivel as the chair moves, automatically aligning the wheels to the direction of movement. Without a caster offset, the vehicle would be omnidirectional (capable of moving in any direction) but still nonholonomic, as the wheels have to be manually aligned to face the direction of desired motion before the vehicle can begin moving. Overall, holonomic drive is preferred for maximum maneuverability.

Our holonomic base uses a powered-caster drive mechanism [28]. It is driven by four motorized casters, and can be thought of as a motorized office chair. The ability to steer all four wheels makes the base omnidirectional, and the caster offset makes the base holonomic, allowing it to instantaneously accelerate in any direction as it does not need to first align the wheels to the direction of motion.

A holonomic mobile base enables easier teleoperation and kinesthetic teaching for collecting imitation learning data. Everyday tasks such as opening doors and cabinets often require sideways motions of the mobile base to improve the workspace of the arm during execution. This useful motion is not immediately available with a differential drive base. Instead, the robot has to replan vehicle trajectories to satisfy nonholonomic constraints, which costs extra motion and time with no added value to the task. A holonomic mobile base, on the other hand, can be much more reactive. It can be moved arbitrarily in any direction no matter the current configuration, allowing an operator to make fine adjustments to the positioning of the base.

A holonomic base is also useful for policy learning and inference. Recent real-robot imitation learning works have converged on the use of position representations, as they are more stable and less

noisy compared to velocities. However, a nonholonomic mobile base can only be controlled in velocity mode [8, 16]. A holonomic base, on the other hand, can be directly commanded to go to a task space position $(x, y, \theta)$ in a repeatable manner, as it can independently control all DOFs with no constraints. In our experiments, we show that we can indeed train high-performing policies for our robot across several mobile manipulation tasks in a real apartment home. Additionally, we show that policies can be learned more easily with data collected from a holonomic base compared to a nonholonomic one.

To facilitate easy data collection with our new mobile manipulator, we also develop a mobile phone teleoperation interface. The interface uses the WebXR API [29] to stream the real-time 6-DoF pose of the mobile phone to a computer, which maps the phone motion to corresponding motions of the mobile base or arm via low-level control. WebXR is supported on most modern Android and iOS phones, so our interface does not require purchase of a separate teleoperation device. In our experiments, we use this teleoperation interface to collect data for training our policies.

Our holonomic mobile base is low-cost ($5–6k USD) and designed from the ground up to optimize for mobile manipulation research productivity. We will fully open source all aspects of this system, including the hardware design, mobile phone teleoperation interface, policy learning setup, and low-level controller. We will also create a documentation webpage for the mobile base, including bill of materials (BOM), a hardware assembly guide with videos, and 3D CAD files. We believe these components can help democratize access to highly maneuverable mobile manipulators, increase ease and practicality of mobile manipulation data collection, and improve research reproducibility by providing a standardized and reusable robot platform.

Our key contributions in this work are thus: (1) an open-source design for a holonomic mobile manipulator, (2) a mobile phone teleoperation interface for easily collecting data with the mobile manipulator, and (3) demonstration that our system is capable of learning policies. Please see our project page at http://tidybot2.github.io for documentation, code, and qualitative videos of our robot in action.

## 2 Related Work

**Mobile manipulation hardware platforms.** Recently, there has been an increased interest in equipping manipulation robots with mobility to demonstrate their full potential for a variety of tasks—especially in domestic settings [8, 30, 31, 32, 33, 34, 35, 36, 37, 38, 39, 40, 41]. These works utilize a number of different mobile bases which we compare to our base in more detail in Tab. 1. For example, several projects use the Tiago mobile manipulator, and learn policies with a combination of RL, simulation, and human motion transfer [42, 43, 44, 30, 45]. Tiago is holonomic through its use of mecanum wheels, but the wheel comes with a few drawbacks. Due to the multiple small rollers on the wheel, the contact point with the ground is discontinuous leading to vibrations during operation. They also provide less traction and have poorer ability to traverse door thresholds and other inevitable household clutter compared to a conventional wheel. Another common category of bases uses differential drive, which prevents them from freely navigating every degree of freedom [46, 36, 8]. An example of such a robot is Everyday Robots' mobile manipulator, which often appears in Google's robot learning papers as a scalable mobile manipulator platform [36, 47, 48]. This robot is neither open source nor purchasable by the public.

While the above works use wheeled bases, several works attach a robot arm to a quadruped and utilize the combined system as a mobile manipulator capable of traversing terrains that wheeled robots cannot access [35, 37, 49, 50]. Though this combination expands the set of navigable environments, these quadrupeds are nonholonomic since they are kinematically constrained by the orientation and placement of their limbs and feet.

**Data collection for mobile manipulation.** To address the lack of robotics data for learning manipulation policies, several works have developed data collection platforms. The majority of these platforms are built for fixed-arm setups [51, 52, 25, 53]. For example, the DROID dataset [25] was collected on a standardized setup with an arm mounted on a portable table. The authors use an

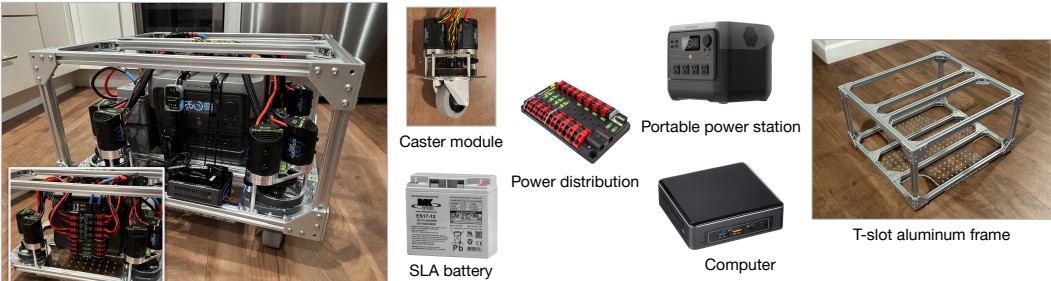

Figure 3: Our mobile base is designed to be modular and easily reconfigurable. It has very few components and can be assembled in 1 to 2 days.

Oculus controller to teleoperate the robot. However, this controller must remain in view of four IR receivers which can lead to unexpected motion if the controller moves out of view and back. Similar to our work, RoboTurk [51, 54] used a mobile phone to teleoperate fixed-base robot arms which is a much more flexible solution and does not require purchasing a dedicated teleoperation device. However, their system suffers from drift as they only rely on IMU measurements and do not use the camera. MART [53] and MOMART [55], which extend RoboTurk to multi-arm and mobile manipulation, respectively, suffer from similar shortcomings and have not been demonstrated on real robots. In our work, we use the WebXR API [29], which combines IMU data with visual odometry based on the phone's camera to mitigate drift. TeleMoMa [45] is a teleoperation framework that supports multiple teleoperation interfaces and three commercially-available, high-cost robots (for a detailed comparison to our low-cost base, see Tab. 1). One of the supported teleoperation devices is a mobile phone app based on ARKit, similar to what we use. Our interface is based on WebXR, which leverages ARKit on iPhone but works on Android as well.

There are several works that propose data collection devices that are hand-held by the human demonstrator [52, 11] but in this case the demonstrator does not get direct feedback on whether a demonstration is kinematically feasible by the robot. Of those approaches, Dobb·E [11] proposes a low-cost reacher-grabber stick with a mounted iPhone to record data. The authors then train visuomotor policies on this data that are deployed on a differential drive Stretch robot [46]. Mobile ALOHA [8] is a dual-arm mobile manipulation platform capable of performing an impressive array of household tasks. However, the robot's differential drive base and large footprint limits its maneuverability, and the arms are not able to reach the ground. Furthermore, the teleoperator is strapped to the back of the platform far away from the end effectors, which can make it hard to teleoperate precise actions. For our system, the teleoperator can freely walk around the scene and get very close when precision is required.

## 3 Hardware Design

We designed this mobile base concept from the ground up to optimize for mobile manipulation research productivity. It is simple, low-cost ($5–6k), and modular (Fig. 3). The core is the drive system, which is based on readily available components from the FIRST Robotics Competition (FRC) [56] ecosystem. A basic frame built out of aluminum T-slot extrusions carries the four motorized caster modules that are powered through a fused power distribution panel by a sealed lead acid (SLA) battery. There are many similar components in the FRC ecosystem that could be used to build similar systems. The large and active community of FRC users and vendors ensures that components are well-documented and readily available.

Our drive system is derived from a set of SDS MK4 swerve modules [57] widely used in FRC. The swerve modules are very similar to a caster, with a wheel that can be actively steered and driven, except they have no caster offset and thus create a nonholonomic base. The modules use two motors with integrated encoders and CAN bus controllers, one for steering and one for driving. Additionally, there is an absolute encoder mounted directly to the steering axis to measure the steer

position, and thereby eliminate the need for a startup homing motion. A USB-to-CAN adapter is used to communicate with the motors and encoders through CAN bus.

We modify the MK4 swerve module to create a holonomic base design by introducing a caster offset using a minimal number of modifications to the stock swerve module: 2 custom 3D-printed wheel mounts and a custom machined shaft. The wheel mounts can be printed with PLA filament on a standard FDM 3D printer, and the shaft can be easily ordered from online machining services like Xometry using our provided CAD file and specifications. All other parts of the off-the-shelf kit are directly reused.

To complete our mobile manipulator, we add a mini PC (Intel NUC) and the Kinova Gen3 arm. Power for compute, manipulation, and peripherals is provided by a high-capacity (768 Wh), fast-charging (0–100% in 70 minutes) portable power station (camping battery) with AC power outlets. The portable power station (8.6 kg) and SLA battery (6 kg) serve as counterweights to prevent the base from tipping over. Note that it would be possible to build electrical circuitry to use only one battery for more space efficiency, but we opted for separate batteries as it provides more flexibility and makes setup much easier.

Our open-source design is highly customizable: the frame can be easily modified to support mounting of different arms or even multiple arms, and many additional sensors such as cameras, microphones, etc., can be easily mounted and powered, to suit the research being done.

## 3.1 Powered-caster vehicle kinematics

For our low-level controller, we model the kinematics of the mobile base largely following the formulation of the powered-caster vehicle (PCV) in Holmberg and Khatib [28], which describes the mapping between the joint space and the operational space $(x, y, \theta)$ of the base.

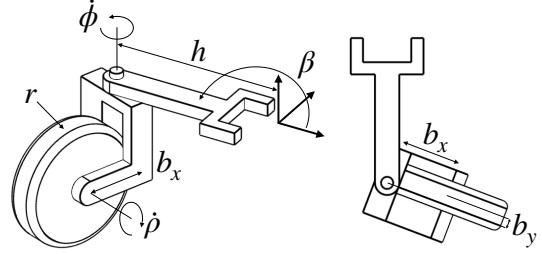

Each caster module is modeled with two revolute joints: a steer joint and a roll joint. The steer joint determines the wheel's steering angle, while the roll joint measures the wheel's rotational movement. We have an incremental encoder in each motor, as well as an absolute encoder on the steer axis of each caster. With these encoder readings, we can calculate the joint positions and velocities using gearbox kinematics.

Figure 4: Isometric and top views of a simplified caster, showing the caster offsets $b_x$ and $b_y$, wheel radius $r$, steer and roll joints $\phi$ and $\rho$, and caster module placement $(h, \beta)$ from the base origin.

The main difference from the original PCV formulation [28] is that instead of a single caster offset $b$, our caster module has a two-dimensional offset: a traditional longitudinal $b_x$ offset as well as a small lateral $b_y$ offset (Fig. 4). This mainly comes as a byproduct of our design criteria to minimize the number of custom parts needed. It would be possible to have no lateral offset $b_y$ by designing more custom parts, or designing a new caster from scratch, but this would significantly lower the accessibility of our design, as creating custom parts requires detailed design work and is very costly to manufacture at low quantities.

## 3.2 Design principles

Our holonomic mobile base is specifically designed and optimized for robot learning research. This objective shapes our key design principles:

**Research flexibility.** We would like users to be able to customize the robot to meet their own specific needs. The frame is designed using standard aluminum T-slot extrusions, providing flexibility to easily adjust the dimensions and shape of the frame. We use a portable power station (camping battery) with four AC power outlets, providing flexibility to power other kinds of robot arms as well as computers of various operating voltages. Components can be simply plugged in directly, with-

Table 1: Mobile base and mobile manipulator comparison

| Specification | Ours | Stretch | Tracer | Ranger Mini | Husky | Fetch | Tiago |
|---|---|---|---|---|---|---|---|
| Holonomic | Yes | No | No | No | No | No | Yes |
| Omnidirectional | Yes | No | No | Yes | No | No | Yes |
| Swappable arm | Yes | No | Yes | Yes | Yes | No | No |
| Footprint (cm) | 50x54 | 33x34 | 57x69 | 50x74 | 67x99 | 51x56 | 54x54 |
| Weight | 34 kg | 24.5 kg | 30 kg | 63 kg | 50 kg | 113 kg | 70 kg |
| Payload | 60 kg | 10 kg | 100 kg | 80 kg | 75 kg | — | — |
| Maximum speed | 1 m/s | — | 1.6 m/s | 1.5 m/s | 1 m/s | 1 m/s | 1 m/s |
| Runtime | 8 h | 2–5 h | 4 h | 7–8 h | 3 h | 9 h | 8–10 h |
| Cost | $5.4k | $25k | $7.6k | $13k | $20k | $100k | $100k |

out needing to set up new circuitry for power supply and voltage conversion. Additionally, we open source the entire control stack, all the way down to low-level control with motor velocity commands. This means researchers can have full control and are not limited to the API functionality exposed by a manufacturer's proprietary software stack.

**Reliable and easily-sourced parts.** One challenge with building a mobile robot for research is the sourcing of parts such as motors, encoders, gears, and custom machined parts, which can be time consuming and costly. We design our drive system mainly using parts sourced from suppliers of the FIRST robotics competition (FRC) [56] for high schoolers. These are the same parts used by over 80,000 competition participants each year. Due to the popularity of the competition, the components are standardized and readily available for purchase from online vendors, and can be ordered online and typically delivered within a week. These parts are very reliable, as they have been battle-tested in the strenuous conditions of competition (125 lb robots moving at high speeds with frequent collisions), and core software components such as CAN drivers, motor control, and battery monitoring are all included. Our caster module design contains only three easily-obtained custom parts as described in Sec. 3.1. The top and bottom plates of the frame are laser-cut acrylic, which can be made with a laser cutter or directly ordered using an online service. All other components of the mobile base can be readily purchased from online retailers such as Amazon.

**Easy assembly and repair.** Our holonomic mobile base design is easy to assemble and can be put together in 1 to 2 days. Putting together the T-slot extrusion frame is the most time-consuming part of the process and takes around 6 hours. Each of the four caster modules can be assembled in less than 30 minutes using only hand tools, and subsequently installed on the frame. The custom wheel mounts for the casters can be 3D printed across 2 days. All electrical wiring can be done in less than 30 minutes and requires no soldering. For repairs, all parts can be easily removed from the frame in a modular fashion, including the caster modules. The robot does not need to be shipped back to a manufacturer when parts break. Instead, replacement parts can be purchased online and directly swapped out for the broken ones.

## 3.3 Specifications

In Tab. 1, we compare our holonomic base with other mobile bases and mobile manipulators commonly used in research. This includes the Stretch mobile manipulator from Hello Robot, the Tracer and Ranger Mini 2.0 AGVs from AgileX, the Husky AGV from Clearpath, and the Fetch and Tiago mobile manipulators. Our mobile base is maximally maneuverable, performant, and flexible, while also having the lowest cost. Note that while the Tiago is holonomic, it uses mecanum wheels which make the robot vibrate as it moves.

**Dimensions.** Our robot has a small footprint (54 x 50 cm), enabling it to navigate in household environments, including narrow doorways and hallways. The top of the mobile base is approximately 37.5 cm above the ground. This height allows the mounted arm to comfortably reach down towards the ground and up towards tabletops and countertops. Note that these dimensions can be easily customized to accommodate other arms.

**Weight.** The mobile base weighs 75 lb (34 kg). We mount a Kinova Gen3 7-DoF arm on top that weighs 27 lb (12 kg) including its mounting plate and power supply unit.

**Payload.** To get a rough idea of the robot's maximum payload, we loaded 270 lb (122 kg) of weight plates onto the mobile base, for a total weight of 345 lb (156 kg). Even with this amount of weight, we found that the motors showed no signs of struggle. For a conservative estimate of a maximum payload suitable for long-term use, we halve the weight and use 60 kg as our estimate. This indicates that our design can comfortably support the weight of many other arms, as long as the frame is redesigned appropriately for securely mounting the arm. Note also that the payload may vary depending on the terrain (we conducted our test on a hard floor).

**Runtime.** We power the robot arm and compute using a portable power station with 768 Wh of capacity, which we found can handle 8 hours of continuous teleoperation runtime. For the motors, since they boot nearly instantly upon connection to power, the SLA battery powering the motors can be easily hotswapped during use if the voltage gets low.

**Traversability.** Our mobile base works on various indoor floor surfaces ranging from hard floor to high pile carpet, and can traverse many common floor obstacles such as door thresholds, floor mats, and elevator gaps. While not intended for outdoor use, we found during transport of our robot that it can handle many outdoor floor obstacles as well, such as bumpy sidewalks, steel construction plates, loading ramps (inclination 6.5 degrees), and speed bumps.

**Odometry.** To evaluate the odometry accuracy of the mobile base, we use motion capture with submillimeter accuracy to measure the pose of the base while driving it in several path shapes. Overall, we find the odometry quality to be quite high, with translation drift of less than 1 cm per meter of distance traveled, and rotation drift of less than 1 deg per 360 deg of rotation. This means that base movements are highly repeatable, as our holonomic base can be controlled in position mode to accurately reach goal poses $(x, y, \theta)$ using odometry feedback.

## 4    Experiments

In these experiments, we aim to show (i) that our teleoperation interface can collect useful demonstration data to successfully train policies for a variety of household mobile manipulation tasks, and (ii) that holonomic drive offers advantages over differential drive for both teleoperation and policy learning. For qualitative videos of autonomous policy rollouts and teleoperation, please see our project page at `http://tidybot2.github.io`.

### 4.1    Imitation learning

We used our phone teleoperation interface to collect demonstrations for the 6 tasks shown in Tab. 2. We collected 100 demonstrations for the shorter *open fridge* task and 50 for all others. Data collection for each task took between 1 and 2 hours for 50 episodes, including overhead for environment resets.

Table 2: Imitation learning results

| Task | Success rate |
|------|:---:|
| Open fridge | 10/10 |
| Wipe countertop | 9/10 |
| Load dishwasher | 7/10 |
| Take out trash | 10/10 |
| Load laundry | 7/10 |
| Water plant | 6/10 |

We then used the data to train a diffusion policy [1] for each task. We trained each policy for 500 epochs and evaluated them by running 10 episodes of policy rollouts. Success rates are shown in Tab. 2. Note that while diffusion policies are typically trained using 200 to 300 demonstrations, we found that 50 was already sufficient for the robot to learn to complete the task successfully. Performance can likely be further improved with more data. These results show that our system is capable of learning high-performing policies for useful tasks in real homes.

### 4.2    Differential drive comparison

To further highlight the advantages of holonomic drive over nonholonomic, we conduct a head-to-head comparison on the *wipe countertop* task. We collect 50 demonstrations of the task with our base

operating in differential drive mode. We implement this by applying the appropriate nonholonomic constraints to the desired base pose and then computing a velocity command to send to the base.

Fig. 5 shows a representative pair of paths from the demonstration data (path begins at the origin), illustrating the greater efficiency enabled by a holonomic base when compared to differential drive. Across all demonstrations for this task, the differential drive base travels on average 4.03 m (average episode duration 65.2 s), whereas the holonomic base travels only 2.03 m (average episode duration 27.4 s).

We then train a diffusion policy using the differential drive data and compare the resulting policy with the holonomic one from before. To ensure a fair comparison, the differential drive policy was trained under identical conditions (50 demonstrations and 500 epochs). While the holonomic base can perform the task with 9/10 success rate, we find that the differential drive policy only achieves 4/10 success rate.

Qualitatively, we find that the differential drive policy does not learn the task as effectively, even though it was given the same number of

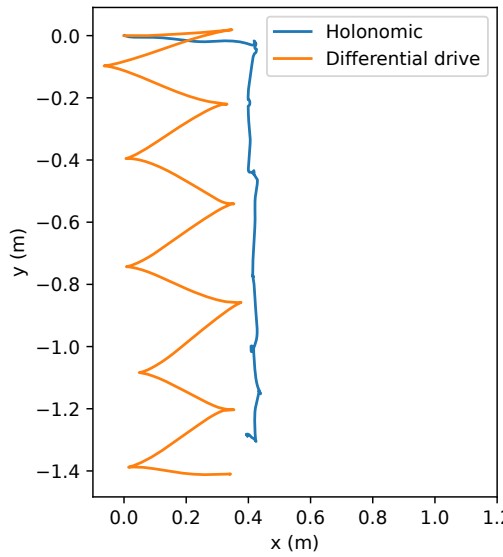

Figure 5: In the *wipe countertop* task, the differential drive robot is forced to take a less efficient path as it is subject to nonholonomic constraints.

demonstrations. The main failure mode is that it frequently skips over portions of the countertop rather than wiping. We believe this is because the learning problem is strictly harder in the case of differential drive. The policy not only has to learn the wiping action, it also has to learn a complicated sideways moving strategy similar to parallel parking, whereas the holonomic policy can directly move the base sideways. Additionally, the differential drive maneuvers cause the camera's view to swerve from side to side, reducing its quality, while the holonomic base can maintain a stable, forward-facing camera view.

## 5 Limitations

One limitation of our mobile base is that it does not backdrive very well due to high steering friction in the caster modules, which can be attributed to the combination of high steer gear ratio (12.8) and small caster offset (14 mm). We confirmed that with the steer gearing removed, the base can indeed backdrive smoothly. The base could be made more easily backdrivable with further mechanical modifications using custom machined parts, but this would come at the cost of reducing the accessibility of the open-source design.

## 6 Conclusion

In this paper, we proposed an open-source design for a mobile manipulator with an inexpensive, robust, and flexible holonomic base. The design uses powered-caster wheel modules, which makes the base holonomic and therefore more maneuverable than commonly used nonholonomic designs. With our easy-to-use mobile phone teleoperation interface, we showed that many household mobile manipulation tasks become easy to demonstrate on our robot. Additionally, we showed that the data collected with our interface can be used to train high-performing policies on a variety of tasks. We hope that the open-source release of this mobile base design and teleoperation interface will enable the robot learning community to easily collect large quantities of mobile manipulation data that will form the basis for policy learning.

**Acknowledgments**

We would like to thank Kevin Zakka, Yixuan Huang, Kevin Lin, Zi-ang Cao, Jingyun Yang, Rika Antonova, Marion Lepert, Sophie Lueth, Haoyu Xiong, Huy Ha, Cheng Chi, Philipp Wu, Fred Shentu, and Zhongke Yi for fruitful technical discussions. We thank Rajat Kumar Jenamani, Rishabh Madan, and the Cornell EmPRISE Lab for open sourcing their compliant controller for the Kinova arm. We would also like to thank Zi-ang Cao, Rika Antonova, and Haoyu Xiong for extensive hardware assistance, as well as Rika Antonova and Zipeng Fu for lending hardware. We are especially grateful towards the FIRST Robotics Competition (FRC) community for developing the rich ecosystem that made this project feasible. For extensive product and logistics support, we would like to thank Mandy Gove, Cory Ness, Omar Zrien, Jacob Caporuscio, and Dalton Smith from CTR Electronics; Ranjit Chahal and Harvey Rico from WestCoast Products; and John Rigsby from Swerve Drive Specialties. The work was supported in part by the Stanford Institute for Human-Centered Artificial Intelligence (HAI), Princeton School of Engineering, the Sloan Foundation, and the National Science Foundation under ECCS-2143601.

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
