# OpenReview forum: "TidyBot++: An Open-Source Holonomic Mobile Manipulator for Robot Learning"
_robot-learning.org/CoRL/2024/Conference — CoRL 2024_

### Official Review · Reviewer_j6ug · 2024-07-19
**Good contribution but need more experiments**

**Originality:** 2
**Technical Quality:** 2
**Clarity Of Presentation:** 4
**Potential Impact:** 3
**Recommendation:** 3
**Confidence:** 5

**Review:**

### Strength
1. The paper makes a clear contribution by providing a low-cost and holonomic platform for robot learning research. The existing mobile platforms are typically non-holonomic, creating difficulty both in data collection and in policy learning, as the tasks become long horizon and the action space becomes non-standard.  The contribution of this paper would greatly lower the barrier for research in mobile manipulation.

2. The supplementary video is well done and showcase the capability of the proposed system

### Weakness

The paper lack experiment and analysis. The current paper only contains one experiment in Figure (should be renamed to Table) 5, which has one user. The paper should provide more details about how the user is selected and why this user, or conduct experiments with a few more users.

Other experiments that are good to have:
* Imitation learning results to justify the claim of the proposed system for robot learning research
* Visualization of the resulting trajectories from a holonomic vs a non-holonomic base

Table 1 should include references.

----
Update after rebuttal: The authors have addressed my concerns with the changes and new experiments. I believe this is a good paper for the conference and have raised my score.

**Quality Of The Limitations Section:**

3

**Questions For Rebuttal:**

How does the proposed system compared to Reachy from Pollen Robotics?

**Robotics Focus:**

4

**Summary Of Paper:**

The paper contributes an open-sourced mobile platform for robot learning research. The two key features of the system are low cost and holonomic, which enables easier and faster task completion.

**Summary Of Recommendation:**

Great contribution but need more experiments. I am willing to raise my score conditioned on more experimental results during the rebuttal.

---

### Official Review · Reviewer_NmwR · 2024-07-22

**Originality:** 3
**Technical Quality:** 3
**Clarity Of Presentation:** 5
**Potential Impact:** 3
**Recommendation:** 3
**Confidence:** 4

**Review:**

This is a very promising paper. The design of the base itself looks quite good and easy to replicate. The holonomic design is a big benefit. The intended use of the base seems to be for imitation learning.

However, the paper doesn't include any imitation learning results. An important criteria for hardware to be useful for imitation learning is to have high repeatability. When teleoperating live, the operator can correct for imprecise hardware in real-time. However, if the hardware dynamics are not repeatable, then the imitation learning policy will fail. For eg. in the chair rearrangement task, the base dynamics need to be accurate, otherwise the robot will not reliably reach the chair. I would like the authors to include a few imitation learning results in their paper. Indeed, prior works that introduce low-cost platforms of this sort typically also include autonomous imitation learning results (eg. GELLO, ALOHA, UMI, etc.).

I would have also liked to see some analysis of the quality of the control and data from the base. For instance, how easy is it to teleoperate by an untrained person? What impact does the caster design have on the repeatability of the actions? What is the quality of the data measured from the base? These analyses are useful because they demonstrate the quality of the hardware and increase the likelihood of adoption (see questions)

**Quality Of The Limitations Section:**

3

**Questions For Rebuttal:**

- What is the action space for the base? Is it $(v_x, v_y, \omega)$?. For a given action command, how does the behavior of the base change for different positions of the caster wheels? For eg. does the base hit singularities for some positions of the caster wheels? How does the imitation learning policy deal with this?
- Similarly, to the above question, if the action space is velocity, how does the robot accelerate / decelerate to the commanded velocity? Does the exact profile depend on the wheel configuration?
- What is the quality of the odometry data from the base?
- Have the authors considered providing a simulation package for the base with accurate dynamics? This could be useful for researchers exploring algorithms in simulation or for sim2real.

**Robotics Focus:**

4

**Summary Of Paper:**

This paper proposes an open source low-cost mobile base built using off-the-shelf parts. The base is a powered caster design and holonomic. There is also an open source mobile app for teleoperation which uses the touchscreen and IMU of the phone. The authors show impressive teleoperation results with their base.

**Summary Of Recommendation:**

I am voting weak reject because of lack of demonstrated downstream application + some missing analysis (see questions). Happy to raise my score if concerns are addressed.

---

### Official Review · Reviewer_rKYA · 2024-07-22
**More scientific questions need to be addressed**

**Originality:** 4
**Technical Quality:** 4
**Clarity Of Presentation:** 4
**Potential Impact:** 4
**Recommendation:** 3
**Confidence:** 5

**Review:**

The contribution of this work is that the author has introduced an open-source platform for a holonomic base that simplifies mobile manipulation, facilitating further research in realistic household robotics applications.

Yes, currently there isn't a suitable mobile manipulation platform for household environments! The author has identified a gap, making a significant contribution to the field.

However, designing a holonomic base is not new. Beyond engineering a base, I believe there are several scientific questions the paper can address:

1. What is the optimal human-robot interface design for teleop a mobile manipulator? Mobile Aloha style? or VR teleop style? Why?

2. How does the holonomic mobile manipulator perform in real-world tasks compared to a differential base with an arm? Are there any tasks that it can perform more efficiently? In which scenarios is a holonomic base particularly necessary?

3. What really matters in Mobile Manipulation Imitation Learning? Can you analyze the results of imitation learning based on varied initial configurations? Which is more effective for a mobile robot with a moving camera, RGB? or point cloud input? I am interested in more learning results instread of just teleop results.

**Quality Of The Limitations Section:**

2

**Questions For Rebuttal:**

I am particularly interested in the following analyses from the author:

Hardware Analysis:

1. Why holonomic? Discuss the advantages of using a holonomic base for mobile manipulation.
2. What are the height limitations of the robot, and how does this affect stability? Are there stability tests, such as tipping over when force is applied?

Software Toolbox:

1. An open-source teleoperation toolkit.
2. Basic simulation environment and URDF files.

I will consider to raise my scores if I see more learning results:

1. Basic ACT/ diffusion policy results with the tasks you picked in the paper.
2. Define the initial distribution for task settings. Success rate metrics. Benchmark tasks for evaluation.

It would be beneficial if the paper discussed:

1. The use of imitation learning for mobile manipulation: Are local or global actions more effective?
2. The choice of sensory input: Is RGB or point cloud data preferable for mobile manipulation tasks?

**Robotics Focus:**

4

**Summary Of Paper:**

The paper introduces a open sourced holonomic base for household mobile manipulation, which is a missing part in robot learning community. The author provided a hardware design and teleop system design.

**Summary Of Recommendation:**

I will consider to raise my scores if I see more learning results.

---

### Decision · Program_Chairs · 2024-09-04

**Decision:**

Accept

**Comment:**

The reviewers found some strengths in this submission, but also clearly articulated some questions for the rebuttal phase.
Thank you for your responses.